# Involvement of service user representatives on a healthcare organizational level at Norwegian Healthy Life Centres: A qualitative study exploring health professionals' experiences

Espen Sagsveen[1]*, Marit By Rise[2], Heidi Westerlund[3], Kjersti Grønning[1,4], Ola Bratås[1]

1 Department of Public Health and Nursing, Faculty of Medicine and Health Sciences, Norwegian University of Science and Technology, Trondheim, Norway, 2 Regional Centre for Child and Youth Mental Health and Child Welfare (RKBU) Central Norway, Department of Mental Health, Faculty of Medicine and Health Sciences, Norwegian University of Science and Technology, Trondheim, Norway, 3 KBT Competence Center for Lived Experience and Service Development, Trondheim, Norway, 4 Nord-Trøndelag Hospital Trust, Levanger, Norway

* espen.sagsveen@ntnu.no

## Abstract

### Background

The involvement of service user representatives in planning, delivering, and evaluating health care services is regarded as essential in Healthy Life Centres (HLCs) to ensure high-quality services. However, information on how HLC-professionals involve service user representatives at a healthcare organizational level at HLCs remains sparse.

### Objective

To explore HLC professionals' experiences involving service user representatives in planning, delivering, and evaluating the HLC services.

### Methods

Five qualitative semi-structured focus group interviews with 27 health professionals from 27 Norwegian HLCs were conducted. Data were analysed using systematic text condensation.

### Results

The involvement of service user representatives at the HLCs varied from well-integrated and systematized to the opposite. The professionals' primary rationale for involving service user representatives was to include the representatives' unique experiential knowledge to ensure the quality of the service. Experiential knowledge was seen as a 'different' competence, which came in addition to professional competence. The professionals' choice of service user representatives depended on the purpose behind the involvement initiative. The HLC professionals often hand-picked former service users according to their health

**Data Availability Statement:** The de-identified data set cannot be shared publicly because of the potential loss of confidentiality. There are ethical and legal restrictions on data sharing due to indirect personal identification, which can be traced back to person, profession, and workplace. It is not possible to publish the original data as participants were guaranteed in the information letter that their interviews would not be publicly available. Therefore, data publication would violate their privacy rights and conflict with the General Data Protection Regulation (GDPR) and The Personal Data Act (national special rules for scientific research). The information letter and study were approved by the Norwegian Centre for Research Data, reference number 43803. This study was a part of the project 'Health Promotion – Worthwhile? Reorienting the Community Health Care Services'. Data could be made available for researchers who meet the criteria for access to confidential data upon request to the project leader, professor Gøril Haugan (email: goril.haugan@ntnu. no).

**Funding:** The Research Council of Norway funded this study under Grant number 238331, "Health Promotion – Worthwhile? Reorienting the Community Health Care Services". The funder had no role in study design, data collection and analysis, decision to publish, or preparation of the manuscript.

**Competing interests:** The authors have declared no competing interests exist.

problems, motivation, and the HLC's need. The professionals said they were responsible for initiating the facilitation to accomplish genuine involvement. Support from their leaders to prioritize these tasks was essential.

## Conclusion

To meet the demand for adequate service user representatives, the HLCs need access to different service user representatives, representing both diagnose-based and generic service user organisations and the public. To achieve genuine involvement, the rationale behind the involvement and the representatives' role must be clarified, both for the HLC professionals and service user representatives. This will require resources for continuous organizational preparation and facilitation.

## Introduction

Service user involvement is regarded as essential in contemporary health care to ensure high-quality health care services [1–5]. As in many other Western countries, a goal in public health policy in Norway is to involve service users in the planning, delivery, and evaluation of health care services [2, 6–10]. In Norwegian health policy documents, service user involvement is presented as a quality issue with the potential to improve services [7, 9, 11, 12]. To achieve this, service users are encouraged to take greater control over their health care and become more involved in developing health care services [8, 9, 13, 14]. Service user involvement has been described as involving service users in clinical settings in their treatment at the direct care level (micro-level), involvement at the healthcare organizational level (meso-level), or society and government level in policymaking arenas (macro-level) [13, 15–17]. In this paper, we explore health professionals' experiences with involving service user representatives at an organizational meso-level following Tritter's [17, p. 276] definition of the concept, 'Ways in which patients can draw on their experience and members of the public can apply their priorities to evaluation, development, organization, and delivery of health services'.

Norway established Healthy Life Centres (HLCs) as a part of a national strategy to promote healthy lifestyles and prevent non-communicable diseases (NCDs) [18]. The HLCs direct their services toward people who need help to change living habits related to, among other things, healthy diet, physical activity, and tobacco cessation [19, 20]. The HLC is an easily accessible primary care service that people can attend through a referral from a general practitioner, other public service personnel, or self-referral [20]. The HLCs offer knowledge-based self-management support for health behavior change through individual and group-based counselling and activities, such as self-management education programs [19, 20]. Intervention periods of 12 weeks are offered, with the possibility to extend the prescription period twice; hence, at some HLCs, a total of 36 weeks of intervention can be experienced [20, 21]. The leading group of professionals employed at HLCs is physiotherapists (approximately 50 percent), followed by nurses, occupational therapists, and clinical dietitians [22].

Professional knowledge and patient experience in self-management education programs should be equally important, implying that health care professionals and service users should cooperate in planning and delivering the programs [23]. The HLCs guidelines emphasize that service users should be actively involved in planning, delivery, and evaluation of services [19,

20] and have a genuine influence on decisions about their treatment, care, and design of services [20].

Approaches to service user involvement at an organizational level take many forms, such as taking part in project groups, steering committees, patient or community advisory boards or councils, governing bodies, and acting as experts in patient education and self-management activities [13, 15, 24–26]. Several studies have shown that involving service users in organizational design and policymaking has the potential to improve safety, quality of care, and health outcomes [2, 16, 24, 27, 28]. However, the literature indicates that involving service user representatives and their experiential knowledge in service planning and design still take place to varying degrees [2, 16, 21, 24, 29]. Research also shows that health professionals are uncertain about conducting service user involvement [10, 16, 30]. Previous studies reveal that despite incentives to strengthen service user involvement in planning and delivering self-management interventions, the interventions are mainly planned and delivered by health professionals without the involvement of service users or lay participants [23, 29, 31, 32]. Service user involvement at an individual level has been explored, showing that service user involvement is essential for the quality of care [30, 33–35]. However, research investigating the involvement of service user representatives in planning, delivering, and evaluating services at an organizational meso-level at HLCs is lacking.

To reach the governments' and HLC guidelines' aim of involvement and offering high-quality health care services that suit patients' needs, service users must be given opportunities to influence the development and delivery of services. So far, little is known about how professionals in HLCs experience and work with service user involvement at an organizational level. Hence, the overall aim of this study was to explore HLC professionals' experiences with involving service user representatives in planning, delivery, and evaluation of the HLC services.

## Materials and methods

This qualitative study used semi-structured focus group interviews to explore professionals' experiences involving service users at Norwegian HLCs. Five interviews with 27 professionals were conducted. Focus group interviews facilitate interaction between the participants and help initiate a recall, applicable when doing an explorative study [36]. The study was conducted in Norway from September 2015 to November 2017.

### Setting and sampling

The sampling was strategic, recruiting participants from HLCs in rural and urban municipalities in Central, South-Eastern, and Southern Norway. The sample of HLCs included well-established and new centres and centres differing in size (regarding the numbers of employees and inhabitants the HLC served). The implementation of interventions varied across the HLCs depending on local policy, resources, and competence. Only participants from HLCs offering a 12-week follow-up period with individual health counselling were included in the study. An additional inclusion criterion was that participants had experiences planning, providing, and evaluating individual- and group-based counselling and activities related to physical training and dietary behavior or tobacco cessation. Participants in focus group interviews number one and two were recruited in collaboration with two local health coordinators. The local health coordinators informed the participants about the study. They sent the names of the participants to the first author (ES), who contacted them with information about the time and place for the interview. The first author recruited participants to focus groups three, four, and five through phone calls. Contact information was found on the websites of the Norwegian Directorate of Health and the municipalities in which the participants were employed. Those

who agreed to participate received an information letter by e-mail, and all signed a written consent form before taking part in the interviews. The study was approved by the Norwegian Centre for Research Data (NSD) (Project no. 43803).

## Data collection

The focus group interviews consisted of four to eight participants in each group, lasting for approximately 90 minutes and taking place at different locations. The interviews with focus groups one and five were conducted in the areas of the HLCs, and meeting two, three, and four took place in meeting rooms located at a university campus and a public community- and conference centre. The focus groups were organized according to the inclusion criteria, with a mix of new and well-established HLCs and participants of different ages and experiences. Practical considerations such as group size, travel distance, and available meeting dates, times, and places convenient for the participants were also considered. All interviews were conducted according to an interview guide developed by the first and fifth authors. The development process included a literature review and discussions with co-authors, one of them with extensive research experience in service user involvement. Open-ended questions were used, inviting the participants to talk about their own experiences. Participants were encouraged to discuss and illustrate their experiences with service user involvement with examples from their practice. See Table 1 for examples of topics from the interview guide.

The interview questions were used as a guide, and the sequence depended on the participants' responses to previous questions.

The first author (ES) operated as the moderator, ensuring that everyone had the chance to express their view. The fifth author (OB) acted as an assistant moderator, making notes throughout the group discussions, and asking supplemental questions at the end of the interviews. All interviews were audio-recorded and transcribed verbatim. After each interview, the first author listened through the digital recording and wrote a summary.

## Data analysis

The data were analysed using systematic text-condensation, a descriptive and explorative method for thematic cross-case analyses [37]. The systematic text condensation procedure consists of four steps [37]. First, all five authors read all transcripts to establish an overview and gain a general impression of the data, searching for preliminary themes related to service user involvement at an organizational level. After reading the transcripts, all authors met to

**Table 1. Main topics in the interview guide.**

| |
|---|
| 1. What do you understand by service user involvement at an organizational level at the HLC? |
| 2. What kind of experiences do you have involving service user representatives?<br>• examples positive and/or negative?<br>• examples of challenges (if so, how)?<br>• has service user involvement ever come into conflict with your professional judgement? / Can service user involvement conflict with your professional judgement (if so, how)? |
| 3. What kind of meaning or significance do you think service user involvement has?<br>• examples of benefit or effect?<br>• influence on the service or on the way you work? |
| 4. What is needed of the HLC or you to achieve successful service user involvement? / How do you facilitate service user involvement?<br>• what are the prerequisites for (successful) service user involvement? |
| 5. How are the perspectives of service user representatives incorporated in the planning, development, and delivery of the services at the HLC? |

discuss preliminary themes. Examples of preliminary themes were: 'Finding the right service user representative', 'Another kind of competence and perspective', and 'Up to date and direct feedback'. In the next step, the first author reviewed the transcripts to develop code groups from the preliminary themes, identifying meaning units, followed by a joint agreement between the authors about the content of the codes.

Further, the first author systematically sorted the meaning units of the actual code groups into a few subgroups exemplifying vital aspects of each code group. Then, the content of every subgroup was reduced into a condensate—an artificial quotation maintaining as far as possible the original terminology used by the participants [37]. After finishing the condensation, illustrative quotations were identified. Finally, the condensed contents were synthesized to generate generalized descriptions and concepts (recontextualized) concerning professionals' experiences involving service user representatives at the HLCs, described as the final themes in the presentation of results. All authors read all interviews to achieve a nuanced perspective on the analysis and possibly reduce single-researcher preconceptions. The research group also validated the interpretations and findings against the initial transcripts to ensure that the synthesized result reflected the original context.

During the analysis, preliminary results were presented and discussed with a research group focused on patient education and user involvement. Preliminary results were also presented and discussed at a national HLC conference focusing on user engagement and a national seminar with service user representatives. Quotes from the transcripts were translated into English by the first author (ES) and then double-checked by the other authors to verify the meaning content. Quotes are used in the result presentation to elaborate and illustrate the findings. Because only one male is represented in the sample, gender is not attached to the citations to anonymize the data.

## Results

Five focus group interviews were conducted with 27 professionals from 27 HLCs from rural and urban municipalities in Norway's mid-, south-eastern, and southern parts. The mean age of the participants was 39.5 years (range 23–66). The largest group of professionals included was physiotherapists, followed by nurses as the second largest. The sample characteristics are presented in Tables 2 and 3.

The results consist of four main themes: (a) variation in the degree of integration of service user involvement; (b) integration of service users' experiential knowledge; (c) choosing the right service user representatives; and (d) the inherent demands in service user involvement. Results are illustrated and elaborated by participant quotes, which are identified with participant (P) code and focus group (FG) number.

### Variation in the degree of integration of service user involvement

The HLC professionals gave very different descriptions of how they involved users at an organizational level. Some described a well-incorporated service user involvement in all parts of their HLC service and that they had worked purposefully with user involvement for some years.

> Our municipality has worked a lot with service user involvement in the last years, and we have systemized it since we have focused on it. We have experts by experience who plan the courses together with us and tell their story at the beginning of the courses and evaluate with us in meetings afterward. In addition to the evaluations we get from the course participants. (P2, FG5)

**Table 2. Demographic characteristics of the participants (N = 27).**

| Characteristics | Number of informants |
|---|---|
| **Gender** | |
| Male | 1 |
| Female | 26 |
| **Age** (mean 39,5, range 23–66) | |
| 18–29 | 8 |
| 30–39 | 6 |
| 40–49 | 8 |
| 50–59 | 3 |
| > 60 | 2 |
| **Profession** | |
| Physiotherapist | 13 |
| Nurse | 8 |
| Educationalist | 2 |
| Clinical dietitian | 2 |
| Occupational therapist | 1 |
| Bachelor's degree public health | 1 |
| **Years of Seniority at the HLC** | |
| < 2 | 8 |
| 2–4 | 12 |
| > 5 | 7 |
| **Percentage of full-time equivalent** | |
| 20–30 | 9 |
| 40–50 | 7 |
| 60–70 | 1 |
| 90–100 | 10 |

**Table 3. Characteristics of the HLCs (N = 27).**

| Characteristics | Number of HLCs |
|---|---|
| **Established** | |
| 2003 or earlier | 2 |
| 2004–2006 | 3 |
| 2007–2009 | 3 |
| 2010–2012 | 9 |
| 2013–2014 | 8 |
| 2015–2016 | 2 |
| **Number of positions** | |
| 1 | 11 |
| 2 | 6 |
| 3 | 6 |
| 4 | 1 |
| 9 | 2 |
| 10 | 1 |
| **Number of inhabitants served** | |
| 1 350–4 999 | 8 |
| 5 000–9 999 | 7 |
| 10 000–19 999 | 6 |
| 20 000–49 999 | 3 |
| 50 000–200 000 | 3 |

Other professionals described less systematized and integrated ways of involving service user representatives. One example was limiting involvement to planning, delivering, and evaluating self-management education courses (learning and mastery courses). However, when describing service user involvement in planning and providing exercise group activities and courses for healthy nutrition, and tobacco cessation, many professionals described the involvement of service user representatives as less systematic and implemented.

> We have focused much on service user involvement, especially at the self-management courses (learning and mastery), since it has been part of a project with other municipalities. We have had close collaboration with some significant service user organisations (patient organisations). So, we have had cooperation with service user representatives. But except for that, we have not been so good at involving them. (P3, FG5)

At the other end of the continuum, some described the involvement of service user representatives as random. While many participants emphasized regular meetings with service user organisations as potentially useful for marketing purposes and adjusting the services, these professionals described that they had not done this.

> It is a good idea to have such a meeting to involve service users at an organizational level. To have a yearly meeting with the major service user organisations (patient organisations) because then we can get our service known at the same time as we get feedback. We have not done that. We are not good at service user involvement at an organizational level. So that would be a great idea. (P3, FG2)

According to the professionals, the service user involvement mainly encompassed collaboration with representatives from service user organisations and present or former HLC service users. Involvement included engaging present and former service users and representatives from service user organisations as co-instructors and discussions partners to plan, deliver, and evaluate service provision, such as exercise groups and self-management education courses.

The professionals' descriptions also showed that the HLCs' size and resources, such as the number of employees, percentage of full-time equivalent, number of inhabitants served by the HLC, and how long the HLC had existed, influenced the degree of involvement of service user representatives. The professionals emphasized collaboration between the HLCs as helpful, e.g., drawing on inspiration from each other and the smaller HLCs 'borrowing' service user representatives from the larger.

> I think we are lucky that we have had a collaboration with the NN municipality. I do not believe that we in the small municipalities had managed service user involvement if we could not use some of their service user representatives. So, I think that collaboration between the municipalities is helpful to develop and provide the HLC-services. (P5, FG5)

## Integration of service users' experiential knowledge

The professionals' primary rationale for involving service user representatives was to include the representatives' unique experiential knowledge to ensure the quality of the service. Experiential knowledge was described as a 'different' competence, which could come in addition to professional competence. The professionals highlighted that they wanted the representatives to bring in new and different perspectives.

> Because it is about gaining the experience that the participants in our services have, probably that's what we understand with service user involvement. The service users have the competence that we as professionals do not have. (P2, FG5)

The professionals expressed that they facilitated learning by using their professional knowledge but that they lacked the unique personal experience of coping with a disease in everyday life. Service user representatives could pass on practical everyday examples that helped service users relate the change of living habits to their own lives.

> And when we look at learning and mastering. Learning is about gaining knowledge about something, and I can talk a lot about that. But mastering a challenge or mastering the disease, I have no experience with that. So, if it is to be a learning and mastering course, I think we must have both perspectives. (P4, FG5)

When the professionals discussed the purpose of involving service user representatives' experiential knowledge in their services, they said that the HLC's 'message' on changing living habits was more trustworthy and better communicated from the representatives. Then, the service users identified themselves with the representatives and their experiences.

> I think when your work is to help people become thinner, and you sit there very thin and well trained, it is easy to say, "you should do this and that". But if someone comes there who has been through a lifestyle change, who can refer to their own experiences, they are more on the same wavelength with that person than they might be with us. (P5, FG5)

Involving service user representatives' experiential knowledge was described as essential in planning and delivering self-management education courses, in contrast to other parts of services, such as courses for increased physical activity and healthy nutrition not defined as self-management education courses. According to the professionals, course participants highly valued the input from service user representatives. As a result, some HLCs had converted all their courses into self-management education courses and always included a service user representative as a co-instructor, sharing their story. The professionals emphasized that the service user representatives' lived experiences strengthened the courses' focus on the participants' health behavior changes.

> There has been a slight shift in the Healthy Life Centres. When we started, it was the healthy life centres concept; it was like going out walking, physical activity was in focus. Mastering and change are much more in focus in all our courses now, for example, related to diet and physical activity, together with service user involvement. Moreover, I think that when you have a common start and end of a course, it is easier for a service user representative to get in. So now we have changed the Healthy life recipe to become a learning and mastering course. (P3, FG5)

The professionals also emphasized continuous evaluation, such as patient satisfaction surveys, as essential to improve the service quality. They conducted the evaluation systematically through standardized appraisal forms and service user meetings, or more informally through ongoing dialogue and reflections with their present service users and co-instructors. The professionals described service user representatives as discussion partners providing essential inputs.

> The feedback you get along the way when you are walking together and have an informal talk is valuable. That is when you often get good thoughts, input, and feedback, which is an indirect evaluation of what works and what does not. (P3, FG1)

## Choosing the right service user representatives

The professionals described the choice of service user representatives as dependent on the purpose behind the involvement initiative, often hand-picking former service users according to their health problem, motivation, and the HLC's need. When searching for a co-instructor to communicate and share experiential knowledge, they chose a service user representative with a positive experience from the HLC courses. When evaluating the services, on the other hand, they preferred to include critical voices with more negative experiences.

> As for those who stand and tell their story, it's very nice to have someone who can tell you that they have made some changes, and "this was challenging and this went well and made it easier", and to give the course participants some good tips along the way. But in the evaluation and the planning, I would like to have the critical voices as well. We learn maybe most from them. It is very nice with those who think we are great, but... (P2, FG5)

To evaluate and get feedback, many professionals preferred recruiting service users who recently had attended an HLC course instead of representatives from a service user organisation. The professionals emphasized that they wanted representatives with first-hand experience who provided direct and spontaneous input from a different perspective.

> I think we probably get the most out of those who have been participants and speak straight from the liver. We have experienced that if you have been a service user representative for too long, you start saying the same thing as us. And that's not what we're interested in. We're interested in getting that supplement. (P2, FG5)

However, the need for more professionalized service user representatives was highlighted as advantageous when planning and developing an overall plan for the municipality's health and care services.

The professionals highlighted a dialogue with the public as essential to developing an HLC service adjusted to each municipality's need for local health-promoting initiatives. Here, choosing representatives from service user organisations was described as insufficient since these organisations represent specific groups of service users, often people with chronic diseases and not non-diagnosed people at risk of developing lifestyle-related diseases. Many professionals described a lack of non-diagnose-specific service user organisations in their municipality or region, making it challenging to find laypeople representing the public. To meet this challenge, one of the HLCs had established a close collaboration with a non-diagnose-specific service user organisation consisting of former HLC service users. This generic organisation constituted the HLC's service user council and participated in planning, delivering, and evaluating the HLC services. When the HLC needed a 'service user voice' to represent the HLC, they used representatives from this organisation.

## The inherent demands in service user involvement

The professionals highlighted that it was essential to achieve what they described as genuine service user involvement with actual participation and co-determination. Some professionals

described experiences of the opposite, service user involvement without any content. One example was when they finished planning the activity before involving the service user representatives. Some said that user involvement had not led to the expected results, either because the service user representatives had been more concerned about another agenda than being representative of the target group or because they had said nothing. In these cases, investing time in user involvement just added workload.

> We have focused on having genuine service user involvement and 'how do we get there?'. I think history in recent years has been full of such 'service user involvement–check', where one either has not listened to the service user representatives or they have not said anything. So, there are some challenges because it takes a bit of effort to find service user representatives who can represent more than themselves. (P2, FG5)

The professionals said they were responsible for initiating the facilitation to accomplish genuine involvement. Close collaboration with the representatives, clarifying roles, and establishing a shared agenda, pleasant surroundings, and relations were examples of necessary facilitation. The professionals described it as challenging to have sufficient time and resources to communicate with the service user organisations and coordinate and follow up the collaboration. To achieve this, leader support to prioritize these tasks was essential. One professional said that even before she would 'open the door' for service user involvement, she made sure she had a plan for finding the time to follow up the collaboration.

> And there is something about that I cannot just say "A", and then think it rolls by itself as it might do with other doors one opens in the context of interdisciplinary work. Then one must also address it, and one must have a plan. So, it takes time. (P3, FG5)

## Discussion and conclusion

### Level of involvement

The professionals in this study highlighted service user representatives' experiential knowledge as unique and essential. This in line with other studies [2, 29, 38, 39]. The professionals' emphasis on experiential knowledge as essential in planning, delivery, and evaluation of health care services to ensure the service quality is in line with the HLCs guidelines, national legislation, and Norwegian health policy, requiring collaboration between service users and professionals [9, 11, 23, 40–42]. However, in line with previous research, our findings show variation regarding how the service user representatives were involved, and the level of involvement [10, 13, 23, 25, 43, 44].

Our findings indicate that when the HLC service is defined and delivered as self-management education programs (learning and mastery courses), the involvement of service user representatives seems to be an integrated part of everyday practice [23, 45]. When the professionals described HLC services as not defined as self-management education programs, however, the involvement of service user representatives was less explicitly and systematically incorporated in everyday practice. The health professionals argued that both professional knowledge and service user experiential knowledge had to be present in self-management education programs to secure the quality of services. The arguments used for including service user expert knowledge align with the ideal described for (Norwegian) learning and mastery courses, that professional expertise and service user experiential knowledge must be present [45]. In line with previous research, our findings show that experiential knowledge reflects lived experiences that professionals cannot capture [44].

Further, our results show that experiential expertise in health care is described, as found elsewhere, as a means that can facilitate patient empowerment leading to improved quality of life and increased quality in health care services [46]. Engaging service user representatives to share experiential knowledge can make group support more relevant and mutually beneficial to participants living with a long-term condition [38]. Our findings align with previous research stating that evidence-based practice lacks relevance and trustworthiness for service users unless it explicitly includes service user expertise [47].

This finding indicates that the recommendations given by the health authorities that health care professionals and service user representatives should cooperate in planning and providing these programs are well known by the HLC professionals. Accordingly, the provision of self-management education programs as learning and mastery courses may facilitate service user involvement becoming more routinely incorporated into the HLC practice.

However, our findings indicate that the involvement often was initiated by the HLC-professionals and took place on the professionals' terms. Previous research has found that health professionals often initiate the involvement [23] and that inviting service user representatives to fit into existing organizational cultures (i.e., to adapt to the organisation's way of working) runs the risk of co-opting a particular group of experienced participants to the exclusion of others [27]. The HLC professionals emphasized that they wanted the service user representatives to bring a different perspective. Researchers argue that if the involvement of service user knowledge becomes embedded in service development and delivery, it may become increasingly professionalized [48–51]. Service users often bring a fresh and naïve perspective, helping professionals reconsider and reframe their assumptions and views [2, 52, 53]. However, research shows that the longer and more closely the service users are involved in a service or project, the more significant the risk that the freshness and naivety will vanish. The service users will often unconsciously take on the assumptions and the views of the professionals and the organisation [29, 51, 54].

In line with previous research, our findings further showed that the health professionals possessed the power to ensure that involvement could occur [55]. Only a few HLCs had established, or formalized service user boards based on a partnership model where service user representatives are involved as co-leaders and have equal representation [2, 56]. The incomplete representation may indicate that many HLCs, despite good intentions, practiced consultative approaches rather than partnership approaches to involve service user representatives [2, 25]. Thus, the service user involvement practiced by some HLCs in this study can be considered consultative and indirect. In the literature, a consultation type of involvement is recognized as a low-level form of involvement [2, 25] and resonates with an indirect and reactive type of involvement described in Tritter's conceptual framework [17]. Indirect involvement is characterized by the service user representatives serving as information suppliers and the professionals gathering information to inform service development and delivery, while the health professionals make the final decisions [17]. Reactive involvement describes the extent to which the service user representatives are responding to a pre-existing agenda or is helping to shape it (proactive) [17]. Hence, our findings indicate a need for more vigorous attention among the HLC professionals upon who is initiating the involvement, and the importance of strategies to facilitate genuine involvement and critical reflections on the nature of the partnerships or collaborations.

## Representation and representativeness

An inherent tension when involving service users at the organizational level is connected to representation and representativeness [27, 49, 54]. Our findings show that the HLC

professionals preferred service user representatives with up-to-date and first-hand knowledge about the HLC-services as former service users. To find these representatives, they hand-picked service user representatives based on health condition, motivation, and the HLC's needs. Previous research shows that when professionals actively select, educate, and socialize certain service users, and those who want to be involved self-select, involvement can become unrepresentative [13, 49, 52]. Articulated and educated service users are more likely to put themselves forward for involvement, reinforcing unrepresentative involvement [27, 49]. Thus, the health professionals should be aware that, like other members of a project and steering group, service user representatives may bring a range of skills and particular biases and limitations [57]. To ascertain whether the interests and concerns of the chosen service user representatives are the same as those not involved, and whether the involvement of the representatives may lead to inequalities, a continuous evaluation of the representativeness is needed [25, 54].

When service users who recently attended an HLC course are chosen as representatives, it's essential to be aware that each individual may simultaneously occupy the role of service user and representative for the public [17, 25]. The service user representatives then enter positions as advisors or consultants and become carriers of service users' perspectives, causes, and concerns as spokespersons for a service user group or organisation [6]. The service user representatives simultaneously entering different positions highlights the importance of clarifying the distinction between individual and collective involvement, and the extent to which the service user representatives act as sole agents or as representatives for a group, community, or population [17, 25]. Thus, clarifying the positions and roles adapted and the chosen perspective becomes essential [57]. Further, to advance the understanding and practice of service user involvement at an organizational level at the HLCs, issues of definition and expectations must be made explicit so that appraisals of outcomes can be fair and meaningful [29]. However, there is a need to balance role definitions with openness to views of service users [52]. Giving direction and purpose to involvement may evolve into an inflexible predomination of the patient role by professionals [52].

Previous research shows that legitimacy and credibility are vital ingredients likely to affect service user representatives' ability to cooperate productively with professionals and influence collective health care choices [51]. The service user representatives' credibility is supported by their personal experience as patients, the provision of a structured preparation meeting, and access to population-based data from their community. Legitimacy is encouraged by recruiting a balanced group of participants and the service user representatives' opportunity to draw from others' experiences [51].

Thus, to ensure credibility and legitimacy, access to different service user representatives representing both diagnose-based and generic service user organisations and the public is needed to meet the HLCs' demand for an adequate service user representative. The present study showed that only a few HLCs had established systematic collaboration with generic (non-diagnose-based service organisations). Authors argue that it is necessary to distinguish between the involvement of service users in the role as a service user or as a representative for the public [6, 57]. Fredriksson and Tritter [57] argue that service users (or patients) and the public have different roles, perspectives, experiences, expectations, and interests and should not be grouped. They say that service users have sectional interests as health service users compared to citizens (laypeople) who engage as a public policy agent reflecting societal interests [57]. Thus, as our findings show, using service users as a substitute for the public may fail to achieve the intended goals and benefits of the involvement [57].

Further, the understanding and measuring of the impact of patient and public involvement can, according to Fredriksson and Tritter [57], only develop by applying a more transparent comprehension of the differences. Different public participation contributions, such as

policymaking, consultative user councils, or participation in service delivery, require different roles for the citizens or service users and other demands related to representativeness, performance, and competence of those involved [11].

Accordingly, continuous organizational preparation and facilitation at the HLCs to clarify for all actors the rationale behind the involvement and the representatives' role and what is expected from them is required [1, 16]. Our findings further indicate that some HLCs still need to develop systems to involve service user representatives from the public in planning and designing services. In our study, examples of such systems were the HLCs' establishment of bodies like service user boards and committees or public hearings or meetings.

## Strengths and limitations

The qualitative focus group design facilitated discussions between stakeholders providing data that possibly would not have been retrieved in individual interviews or surveys. The participants representing 27 different HLCs varied in age, working experience, and occupational background, representing well- and newly established HLCs from urban and rural municipalities from different regions across Norway. These variations capture a breadth of experiences, preventing the influence of the culture of one HLC and increase the transferability of the findings [36, 58]. The participants took an active part in the discussions by sharing personal experiences and reflections, and both immediate and profound reflections were captured. Some participants knew each other, but only a few worked together daily. The participants' familiarity with each other may have made them feel comfortable and helped them open up for sharing their personal experiences, preventing one particular group of participants' views from dominating the results.

On the other hand, the participants' experience of knowing each other and working together may have contributed to homogeneity in experiences and opinions, giving uniform and one-sided information, reducing the findings' transferability [36, 59]. The discussions did not reveal any apparent disagreements about the topic. However, the participants had different experiences with the involvement of service user representatives due to organizational structures, such as the size of the HLC, low employment fraction, and inhabitants in their municipality, which also strengthened the transferability of the findings.

Twenty-three of the twenty-seven HLCs included in this study are also part of the data collection for the study in reference 30, which explored how HLC professionals experienced service user involvement at an individual level. The analysis of this study aimed to explore HLC professionals' experiences involving service user representatives in planning, delivering, and evaluating HLC services at a healthcare organizational level. In the analysis, the research group was conscious of separating findings that dealt with the individual and the organizational (system) level, not presenting findings already included in reference 30.

Another strength is that a group of different researchers conducted the analysis and writing of the paper. The composition of the group helped ensure the reliability of the findings. The research group consisted of persons with different professional backgrounds and practical experiences from nursing, health sciences, psychology, and public and mental health to include diverse perspectives. Notably, the third author (HW) has extensive experience as a public representative working with service user involvement. The contribution of different perspectives and the rigorous analytical process helped strengthen the reliability and accuracy of the findings [60]. Furthermore, the first author cross-checked the interpretations and findings against the initial transcript during the various analysis steps to strengthen the validity of the analysis and results, ensuring that the findings are derived from the data [37]. In addition, to enhance the trustworthiness of the findings, the interpretations were critically discussed between the

co-authors with multiple backgrounds and in a research group of experienced researchers to reduce the possibility that preconceptions affected the interpretation.

Only one male HLC professional participated in the study. While no official statistics describe the gender distribution among HLC staff, a national summary of contact persons in HLCs shows that approximately 10% are male, reflecting the sample in this study. A possible limitation was that no service user representatives were interviewed. Including practice-based knowledge through the perspective of the service user representatives could have added another perspective and thereby strengthened the study.

## Conclusion and implications

Many HLCs still need to develop systems to involve service user representatives from the public in more direct and proactive ways in planning and developing services. To meet the HLCs' demand for adequate service user representatives, the HLCs' need access to different types of service user representatives, representing both diagnose-based and generic service user organisations and the public. Hence, it becomes essential to clarify the role adapted and the perspective chosen and simultaneously balancing the definition of role with openness to the service user representatives' views avoiding inflexible predomination of the patient role by the professionals. This implies a continuous evaluation and facilitation to ascertain whether the interests and concerns of the chosen service user representatives are the same as those not involved. To advance the understanding and practice of service user involvement at an organizational level at the HLCs, issues of definition and expectations must be made explicit so that appraisals of outcomes can be fair and meaningful. To achieve genuine involvement resources should be invested in continuous organizational preparation and facilitation clarifying for all actors the rationale behind the involvement, the representatives'role, and what is expected from them.

## Acknowledgments

This study was a part of the project 'Health Promotion–Worthwhile? Reorienting the Community Health Care Services'. We would like to thank the project leaders, professor Gørill Haugan and professor Toril Rannestad. A special thanks to professor Rannestad for her valuable professional input and constructive discussions. Thanks also to KBT Competence Center for Lived Experience and Service Development. Finally, we want to thank the informants who generously contributed with their time and expertise.

## Author Contributions

**Conceptualization:** Espen Sagsveen, Ola Bratås.

**Data curation:** Espen Sagsveen, Ola Bratås.

**Formal analysis:** Espen Sagsveen, Marit By Rise, Kjersti Grønning, Ola Bratås.

**Funding acquisition:** Ola Bratås.

**Investigation:** Espen Sagsveen.

**Methodology:** Espen Sagsveen, Marit By Rise, Kjersti Grønning.

**Project administration:** Ola Bratås.

**Supervision:** Marit By Rise, Kjersti Grønning, Ola Bratås.

**Validation:** Espen Sagsveen, Marit By Rise, Heidi Westerlund, Kjersti Grønning, Ola Bratås.

**Writing – original draft:** Espen Sagsveen, Marit By Rise, Heidi Westerlund, Kjersti Grønning, Ola Bratås.

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
