## [Decision Letter · Decision Letter 0]

8 Nov 2022

PONE-D-22-06962Involvement of service user representatives on a healthcare organizational level at Norwegian Healthy Life Centres: A qualitative study exploring health professionals’ experiences.PLOS ONE

Dear Dr. Sagsveen,

Thank you for submitting your manuscript to PLOS ONE. After careful consideration, we feel that it has merit but does not fully meet PLOS ONE’s publication criteria as it currently stands. Therefore, we invite you to submit a revised version of the manuscript that addresses the points raised during the review process.

We look forward to receiving your revised manuscript.

Kind regards,

Ayse Ulgen, PhD, MGM

Academic Editor

PLOS ONE

Journal Requirements:

Reviewers' comments:

Reviewer's Responses to Questions

**Comments to the Author**

1. Is the manuscript technically sound, and do the data support the conclusions?

Reviewer #1: No

Reviewer #2: Yes

2. Has the statistical analysis been performed appropriately and rigorously? 

Reviewer #1: N/A

Reviewer #2: N/A

3. Have the authors made all data underlying the findings in their manuscript fully available?

Reviewer #1: No

Reviewer #2: Yes

4. Is the manuscript presented in an intelligible fashion and written in standard English?

Reviewer #1: Yes

Reviewer #2: Yes

5. Review Comments to the Author

Reviewer #1: The manuscript deals with an important question, whether the inclusion of individuals who have used Healthy Life Centers (HLCs), a.k.a service user representatives, improve the quality of care at the HLCs when these representatives are participate in the decision making process. By participate, I mean being part of the development and delivery of services (authors' own writing, page 5, line 108). The study was performed using data from Norway HLCs in the time frame 2015 to 2017.

From a practical standpoint, this study makes sense. Service user representatives have a unique perspective, having been treated at a HLC. Also, there is the potential for bias reduction, since professionals have a stake in presenting any results in a positive fashion.

I found the four main themes in the Results section the most helpful in terms of specific examples to test the authors' hypothesis. Of particular help were the quotes. They provided real information regarding HLC practices, and insights from the professionals themselves.

The conclusion was interesting, in that the authors report that many of the HLCs have not implemented plans to make use of service user representatives (page 24, line 571).

This manuscript reads like it is written for those in the specific field of the researchers. While the writing was of good quality in terms of grammar and sentence structure, most of the terms in the text were either not defined, or defined without specific examples. Therefore, it limits the scope of readers.

An example regarding definitions is the term service user representative. It is used in the title (page 1, line 2) and in the very first sentence of page 2 (line 28). It appears that the definition may be found in references (1-5), provided on page 3 (line 64). However, this term is so important, so central to the manuscript, that a definition with examples should be given. Otherwise we are left to wonder, is such a representative someone who, for example, has undergone surgery and has stayed overnight at the HLC, or has received outpatient surgery (e.g., for cubital tunnel syndrome), or has simply come for a wellness checkup?

Another term is HLC. It may be that an HLC means something quite specific in Norway, so only a subset (maybe none) of the procedures listed above are performed. However, it is almost certain that HLC may mean something quite different in other countries. That is, what services do HLCs in Norway provide? Do they differ by region? If so, how does one make general conclusions?

A third example is professional. Does this term refer to doctors, physician-assistants, social workers, housekeeping staff, administrators, all of the above, some of the above?

It is critical to have these terms defined. Without the definitions, we cannot understand what research is being done.

The sample size is quite small (27), and so it is unclear why a method like "systematic text condensation" (page 7, line 165), that sounds quite intricate, was needed to glean information. One would imagine that all information could be gathered just be reading the text from the focus groups.

The authors report that the interview guide used was written by the first and fifth author (page 7, line 150). No citation is given there. I am curious as to whether any review of the robustness of the interview guide has been performed. As a perhaps extreme example, a review of the Diagnostic and Statistical Manual of Mental Disorders, Fifth Edition, was published in 2013 (Regier et al., The DSM-5: Classification and criteria changes, World Psychiatry. 2013 Jun; 12(2): 92–98).

In general, I think the caliber of this manuscript could be improved by "front loading" the manuscript with specific definitions and examples for all the major terms.

Reviewer #2: GeneraLcomments: The paper addresses an important topic concerning how professionals execute their work in relation to the public or users. In some instances, user involvement is necessary, or constitutes an improvement to the professional services delivered.

The paper seems to be a direct continuation of the study described in ref 28. This should be clarified, as the two papers’ description of methods and data collection are very similar, and it may seem like some of the same HLCs were included in the data collection. Please discuss this and what impact use of the same informants, if this was the case, would mean to the findings.

Methods:

P6, line 138: Was an ethical approval provided for the study?

P7: Please name, if it was used, the software used for text analysis.

P7, line 167: Do you mean “all five authors”? Or just “four authors”?

Results:

P23, Lines 559-561: Delete, this is a repetition.

6. PLOS authors have the option to publish the peer review history of their article (what does this mean?). If published, this will include your full peer review and any attached files.

Reviewer #1: **Yes: **Derek Gordon

Reviewer #2: **Yes: **Øydis Ueland

---

## [Author Response · Author response to Decision Letter 0]

29 Jan 2023

Response to the Reviewer Comments

Reviewer #1: The manuscript deals with an important question, whether the inclusion of individuals who have used Healthy Life Centers (HLCs), a.k.a service user representatives, improve the quality of care at the HLCs when these representatives are participate in the decision making process. By participate, I mean being part of the development and delivery of services (authors' own writing, page 5, line 108). The study was performed using data from Norway HLCs in the time frame 2015 to 2017.

From a practical standpoint, this study makes sense. Service user representatives have a unique perspective, having been treated at a HLC. Also, there is the potential for bias reduction, since professionals have a stake in presenting any results in a positive fashion.

I found the four main themes in the Results section the most helpful in terms of specific examples to test the authors' hypothesis. Of particular help were the quotes. They provided real information regarding HLC practices, and insights from the professionals themselves.

The conclusion was interesting, in that the authors report that many of the HLCs have not implemented plans to make use of service user representatives (page 24, line 571).

This manuscript reads like it is written for those in the specific field of the researchers. While the writing was of good quality in terms of grammar and sentence structure, most of the terms in the text were either not defined, or defined without specific examples. Therefore, it limits the scope of readers.

An example regarding definitions is the term service user representative. It is used in the title (page 1, line 2) and in the very first sentence of page 2 (line 28). It appears that the definition may be found in references (1-5), provided on page 3 (line 64). However, this term is so important, so central to the manuscript, that a definition with examples should be given. Otherwise we are left to wonder, is such a representative someone who, for example, has undergone surgery and has stayed overnight at the HLC, or has received outpatient surgery (e.g., for cubital tunnel syndrome), or has simply come for a wellness checkup?

Response: We have made a minor rewriting in the introduction section to clarify the concept of service user representative (page 4, line 74 and line 93). 

This article explores the concept of service user representatives from the professionals` perspective. Therefore, we consciously chose not to give a further definition with examples of the concept than the one found in Tritter`s definition reproduced in the introduction (page 4, lines 75-77).

Another term is HLC. It may be that an HLC means something quite specific in Norway, so only a subset (maybe none) of the procedures listed above are performed. However, it is almost certain that HLC may mean something quite different in other countries. That is, what services do HLCs in Norway provide? Do they differ by region? If so, how does one make general conclusions?

Response: We have rewritten the introduction describing Healthy Life Centres (HLCs) with more details regarding the service provided (page 4, lines 80-84 and 87-89).

We have added a sentence about the HLCs variation regarding the implementation of interventions depending on local policy, resources, and competence (page 6, lines 135-137). 

These variations and their implications for the involvement of service user representatives are included as a part of our discussion (pages 11-12, lines 264-273).

A third example is professional. Does this term refer to doctors, physician-assistants, social workers, housekeeping staff, administrators, all of the above, some of the above?

Response: We have added information about groups of professionals employed at the HLCs in the introduction (page 4, lines 89-91) and in the results section (page 9, line 214).

It is critical to have these terms defined. Without the definitions, we cannot understand what research is being done.

The sample size is quite small (27), and so it is unclear why a method like "systematic text condensation" (page 7, line 165), that sounds quite intricate, was needed to glean information. One would imagine that all information could be gathered just be reading the text from the focus groups.

Response: We chose systematic text condensation because it is a descriptive and explorative method for thematic cross-case analysis of different types of qualitative data, such as interview studies (focus groups). Our goal was to conduct in-depth analyses exploring the interview data thoroughly rather than just touching a surface level.

We see systematic text condensation as a valuable strategy for this purpose and as a analyse method developed from traditions shared by most methods for qualitative data analysis. The method offers a process of intersubjectivity, reflexivity, and feasibility while maintaining a responsible level of methodological rigour.

The authors report that the interview guide used was written by the first and fifth author (page 7, line 150). No citation is given there. I am curious as to whether any review of the robustness of the interview guide has been performed. As a perhaps extreme example, a review of the Diagnostic and Statistical Manual of Mental Disorders, Fifth Edition, was published in 2013 (Regier et al., The DSM-5: Classification and criteria changes, World Psychiatry. 2013 Jun; 12(2): 92–98).

Response: The interview guide was made explicit for this study to explore service user involvement at an organizational level in the Healthy Life Centres context. The rationale behind the study was a need for more knowledge about the involvement of service user representatives in the HLCs.

To strengthen the accuracy of the interview guide, the development of the question included a literature review and discussions with co-authors, one with extensive research experience on service user involvement and one with extensive experience as a service user representative. In addition, after each focus group interview, the research group reviewed the interview guide, and minor adjustments were made.

In general, I think the caliber of this manuscript could be improved by "front loading" the manuscript with specific definitions and examples for all the major terms.

Reviewer #2: 

General comments: The paper addresses an important topic concerning how professionals execute their work in relation to the public or users. In some instances, user involvement is necessary, or constitutes an improvement to the professional services delivered.

The paper seems to be a direct continuation of the study described in ref 28. This should be clarified, as the two papers’ description of methods and data collection are very similar, and it may seem like some of the same HLCs were included in the data collection. Please discuss this and what impact use of the same informants, if this was the case, would mean to the findings.

Response: As you point out, some HLCs are part of both studies. A total of 27 HLCs are included in this study. Twenty-three of these are also included in the data collection for the study in reference 30 (in the Track changes file), which explored how HLC professionals experienced service user involvement at an individual level. The interview guide used for the four first focus group interviews contained questions aimed at service user involvement, both at the individual and the organizational (system) levels. Focus group five contained only questions investigating service user involvement at the organizational level. 

The analysis of this study aimed to explore service user involvement at the organizational level. In the analysis, the research group was conscious of separating findings that dealt with the individual and the organizational (system) level, not presenting findings already included in the article in reference 30. We intended to present two studies exploring service user involvement at different levels.

We have added some text to clarify the connection between the two studies (page 23, lines 561-567).

Comment - Methods:

P6, line 138: Was an ethical approval provided for the study?

Response: Thank you for informing us that the sentence about the ethical approval provided for the study was missing in the main manuscript. We have added the information under the setting and sampling section (page 7, line 151). 

The information about the ethical approval can also be found in the Ethics Statement in the manuscript PDF file.

P7: Please name, if it was used, the software used for text analysis.

Response: No software was used for text analysis.

P7, line 167: Do you mean “all five authors”? Or just “four authors”? 

Response: Thank you for bringing the error to our attention. The correct is all five authors (page 8, line 181). 

Comment - Results:

P23, Lines 559-561: Delete, this is a repetition.

Response: Thank you for bringing the error to our attention. The repetition has been deleted (page 24, line 575).

Response Data Availability statement

The de-identified data set cannot be shared publicly because of the potential loss of confidentiality. There are ethical and legal restrictions on data sharing due to indirect personal identification, which can be traced back to person, profession, and workplace. It is not possible to publish the original data as participants were guaranteed in the information letter that their interviews would not be publicly available. Therefore, data publication would violate their privacy rights and conflict with the General Data Protection Regulation (GDPR) and The Personal Data Act (national special rules for scientific research). The information letter and study were approved by the Norwegian Centre for Research Data, reference number 43803.

This study was a part of the project ‘Health Promotion – Worthwhile? Reorienting the Community Health Care Services’. Data could be made available for researchers who meet the criteria for access to confidential data upon request to the project leader, professor Gøril Haugan (email: goril.haugan@ntnu.no).

---

## [Decision Letter · Decision Letter 1]

21 Jul 2023

Involvement of service user representatives on a healthcare organizational level at Norwegian Healthy Life Centres: A qualitative study exploring health professionals’ experiences.

PONE-D-22-06962R1

Dear Dr. Sagsveen,

We’re pleased to inform you that your manuscript has been judged scientifically suitable for publication and will be formally accepted for publication once it meets all outstanding technical requirements.

Kind regards,

Ayse Ulgen, PhD, MGM

Academic Editor

PLOS ONE

Reviewers' comments:

Reviewer's Responses to Questions

**Comments to the Author**

1. If the authors have adequately addressed your comments raised in a previous round of review and you feel that this manuscript is now acceptable for publication, you may indicate that here to bypass the “Comments to the Author” section, enter your conflict of interest statement in the “Confidential to Editor” section, and submit your "Accept" recommendation.

Reviewer #1: All comments have been addressed

Reviewer #2: All comments have been addressed

2. Is the manuscript technically sound, and do the data support the conclusions?

Reviewer #1: Partly

Reviewer #2: Yes

3. Has the statistical analysis been performed appropriately and rigorously? 

Reviewer #1: I Don't Know

Reviewer #2: N/A

4. Have the authors made all data underlying the findings in their manuscript fully available?

Reviewer #1: Yes

Reviewer #2: Yes

5. Is the manuscript presented in an intelligible fashion and written in standard English?

Reviewer #1: Yes

Reviewer #2: Yes

6. Review Comments to the Author

Reviewer #1: (No Response)

Reviewer #2: L 561-567: In the Strength and limitations part, should the ref (28) in the new text be changed to ref 30.

7. PLOS authors have the option to publish the peer review history of their article (what does this mean?). If published, this will include your full peer review and any attached files.

Reviewer #1: No

Reviewer #2: No

---

## [Editor Report · Acceptance letter]

26 Jul 2023

PONE-D-22-06962R1 

Involvement of service user representatives on a healthcare organizational level at Norwegian Healthy Life Centres: A qualitative study exploring health professionals’ experiences. 

Dear Dr. Sagsveen:

I'm pleased to inform you that your manuscript has been deemed suitable for publication in PLOS ONE. Congratulations! Your manuscript is now with our production department. 

Kind regards, 

on behalf of

Dr. Ayse Ulgen 

Academic Editor

PLOS ONE